# Similarity Study of Electromagnetic and Underwater Acoustic Scattering by Three-Dimensional Targets in Unbounded Space

**Jie Wang** [1,*] , **Hai Lin** [1] , **Huaihai Guo** [2] , **Qi Zhang** [2] and **Junxiang Ge** [1,3,*]

1   School of Electronics and Information, Nanjing University of Information Science and Technology, Nanjing 210044, China
2   School of Marine Sciences, Nanjing University of Information Science and Technology, Nanjing 210044, China
3   Institute of Electronics Information Technology and System, Nanjing University of Information Science and Technology, Nanjing 210044, China
*   Correspondence: henuwangjie@163.com (J.W.); jxge@nuist.edu.cn (J.G.)

**Abstract:** The characterization of targets by electromagnetic (EM) scattering and underwater acoustic scattering is an important object of research in these two related fields. However, there are some difficulties in the simulation and measurement of the scattering by large targets. Consequently, a similarity study between acoustic and EM scattering may help to share results between one domain and the other and even provide a general reference method for the simulation of scattering characteristics in both fields. Based on the method of physical optics, the similarity between the EM scattering of conductors and the acoustic scattering of soft/hard targets and the similarity between the EM scattering of dielectrics and the acoustic scattering of elastics are studied. In particular, we derive how to transfer quantities from one domain into another so that similar scattering patterns arise. Then, according to these transfer rules, the EM scattering and acoustic scattering of three typical targets with different types of boundaries were simulated and measured, and the simulated EM scattering and acoustic scattering curves were found to be in perfect agreement, with correlation coefficients above 0.93. The correlation coefficients between the electromagnetic and acoustic scattering patterns were above 0.98, 0.91, and 0.65 for three typical targets. The simulated and measured scattering results verify the proposed similarity theory of EM and acoustic scattering, including the transfer from one domain into the other and the conditions of EM and acoustic scattering, and illustrate that the acoustic scattering characteristic of the target can be simulated using the EM scattering based on the derived conditions and vice versa.

**Keywords:** acoustic scattering; electromagnetic scattering; similarity analysis; mutual simulation; correlation coefficients

## 1. Introduction

In the study of EM scattering of targets, many laboratory capacities and types of software can be used, but inherent difficulties and shortcomings still remain, for example, the generation of EM wave signals with ultrashort pulses of a few periods and the high space requirements of large targets [1,2]. The speed of EM waves is so fast that higher difficulty and costs are demanded in high-distance resolution measurements. However, the acoustic speed is relatively slow, and ultra-short pulses are easier to achieve, which is very beneficial in the analysis of high-resolution scattering characteristics [3,4]. Compared with the EM wave, the frequency of the acoustic wave is much lower at the same wavelength. However, there are also many limitations to the research on acoustic scattering, such as software, fewer large anechoic tanks, and inconvenient test environments.

EM waves and acoustic waves are two different types of waves. An EM wave is a non-mechanically transverse wave (vector wave), and an acoustic wave is a mechanical kind of wave. The analogy of the physical parameters, impedance characteristics, propagation characteristics, and reflectivity of EM waves and acoustic waves are analyzed in detail

in [5,6]. In [7,8], the acoustic–EM analogy for the reflection–refraction problem is analyzed in detail, and the EM reflection–refraction problem is solved by using the analogy between cross-plane shear waves in the symmetry plane of a monoclinic medium and a transverse-magnetic wave.

In addition, the study of the influence of acoustic waveguides on the transmission and scattering of acoustic waves is very important in underwater acoustic research [9–11]. Correspondingly, similar waveguide phenomena also exist in the propagation of electromagnetic waves in the atmosphere [12,13] and have been successfully applied in long-range radar and communications [14].

Because of their similar characteristics, similar theories and techniques have been applied in both EM-wave and acoustic-wave scattering studies, such as metamaterial technology, phased array, multi-beam, scattering characteristics, and imaging technology [15–20]. In [15], the T-matrix is adopted for the acoustic and EM scattering of circular disks. In [17], the scattering relations of acoustic and EM waves for point sources are analyzed, and general scattering theorems are proved. A sawtooth structure is widely used to weaken the edge diffraction of the EM wave; similarly, the sawtooth structure can also be used to reduce the edge diffraction of acoustic waves [21,22]. Consistent geometric diffraction theory was proposed by Keller and has been widely used in the simulation of the edge diffraction of acoustic and EM waves [23–25]. Electromagnetic-inspired acoustic metamaterials are studied in [26], with a focus on their propagation, radiation, and reflection. The application of surface acoustic wave filters in microwave communication systems demonstrates similar physical characteristics of EM waves and acoustic waves [27].

In [28], simulated high-resolution acoustic data and radar range profiles are studied, and many merits of simulated underwater acoustic data are introduced in detail. The authors introduce the conversion between EM scattering by a perfectly conducting sphere and acoustic scattering by a rigid sphere [29]. Further, in [30], the conversion and scaling relationships between the scattering of EM waves by a dielectric cylinder and the scattering of the acoustic wave by a non-shear-stress cylinder are studied. Even the targets selected in [29,30] are too simple: one is a standard ball, and the other is a 2D cylinder. Importantly, the above studies of EM scattering and acoustic scattering show similar characteristics of EM and acoustic scattering.

Overall, similarity studies of EM and acoustic scattering are expected to provide a new reference analysis method for underwater (or air) unmanned aerial vehicles, ships, aircraft, submarines, and other large types of equipment. However, there are still some unsolved problems in the similarity study of EM scattering and acoustic scattering. Examples include the similarity of EM scattering by 3D conductor/dielectric targets to acoustic scattering by 3D soft/hard/elastic targets.

Based on the method of physical optics, this work studies the similarity between EM scattering by conductors and acoustic scattering by soft/hard targets, as well as the similarity between EM scattering by dielectrics and acoustic scattering by elastics. Meanwhile, conditions for scattering similarity relationships are proposed. Then, according to the similarity conditions, the EM scattering and acoustic scattering of three typical targets with various boundaries are simulated and measured. The results verify the proposed similarity theory of EM and acoustic scattering and illustrate that the acoustic scattering characteristics of the target can be simulated by EM scattering under certain boundary conditions and vice versa.

## 2. Similarity of EM Scattering by Conductors and Acoustic Scattering by Hard Targets

In the research on and measurement of EM scattering and acoustic scattering, targets with a size of $D > 10\lambda$ (or $k \cdot D > 10$) are usually treated as large targets [31], where $k$ is the wave number and $D$ is the aperture of targets. For large targets, as the size increases, it better approximates the scattering by physical optics; the difference between the two kinds of scattering will gradually decrease, and good agreement can be obtained in an ideal state. In this section, based on the method of physical optics, EM scattering from 3D conductor

targets and acoustic scattering of hard/soft targets are studied and analyzed. Then, the similarity relationship is determined, and the constraint of the similarity of EM scattering and acoustic scattering is proposed.

### 2.1. EM Scattering of Conductors

Based on the Stratton–Chu equation, as shown in Figure 1, the scattering electric field $\vec{E}^s$ and the magnetic field $\vec{H}^s$ can be expressed as [31,32]

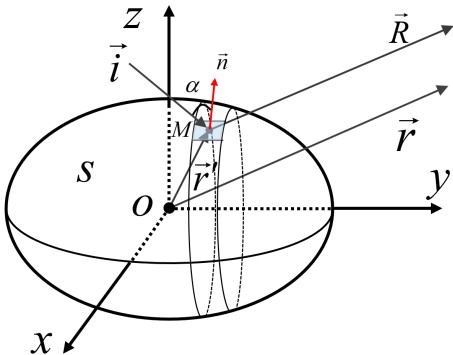

**Figure 1.** Schematic diagram of acoustic wave incident on a target.

$$\vec{E}^s = \iint_s \left[ i\omega\mu \left( \vec{n} \times \vec{H} \right) \cdot G + \left( \vec{n} \times \vec{E} \right) \times \nabla G + \left( \vec{n} \cdot \vec{E} \right) \nabla G \right] ds \tag{1}$$

$$\vec{H}^s = - \iint_s \left[ i\omega\mu \left( \vec{n} \times \vec{E} \right) \cdot G + \left( \vec{n} \times \vec{H} \right) \times \nabla G + \left( \vec{n} \cdot \vec{H} \right) \nabla G \right] ds \tag{2}$$

where $G = e^{-ikr}/4\pi r$ is the Green's function in free space, $r$ is the distance from the objects, $\vec{s} = \vec{r}/r$ is the unit vector of the scattered field direction, and $\vec{n}$ is the outward-directed normal at $ds$. $\vec{E}$ and $\vec{H}$ are the total electric field and magnetic field on the boundary, respectively. In the far field $r \rightarrow \infty$ and $\nabla G \approx ik\vec{s}G$, the scattered electric and magnetic field is perpendicular to $\nabla G$, and there are no components in the scattering direction of the electric and magnetic fields on the conductor surface. Therefore, the third item in the integral equation is zero [31]. The Stratton–Chu equation is expressed as

$$\vec{E}^s = ikG \iint_s \vec{s} \times \left[ \left( \vec{n} \times \vec{E} \right) - \eta\vec{s} \times \left( \vec{n} \times \vec{H} \right) \right] e^{ik\vec{r}\cdot(\vec{i}-\vec{s})} ds \tag{3}$$

$$\vec{H}^s = ikG \iint_s \vec{s} \times \left[ \left( \vec{n} \times \vec{H} \right) - Y\vec{s} \times \left( \vec{n} \times \vec{E} \right) \right] e^{ik\vec{r}\cdot(\vec{i}-\vec{s})} ds \tag{4}$$

where $\vec{i}$ and $\vec{s}$ are the direction of the incident and scattered wave, $\eta$ and $Y$ are the impedance and conductance of the free space, and $\eta\vec{H}^s = \vec{s} \times \vec{E}^s$. In the far field, the $R$ is the distance between the receiver and scattering point M, $R = \left| \vec{R} \right| = \left| \vec{r} - \vec{r'} \right| \approx r - \vec{r'} \cdot \vec{s}$, and $\vec{r'}$ is the location vector of M. Therefore, the scattered field can be calculated using either of the above two equations, and the electric field scattering equation is used in the following introduction.

For the conductors, $\vec{n} \times \vec{E} = 0$ and $\vec{n} \times \vec{H} = 2\vec{n} \times \vec{H}_i = \vec{J}_s$, where $\vec{H}_i$ is the incident magnetic field and $\vec{J}_s$ is the current density. Therefore, Equation (3) can be written as

$$\vec{E}^s = -2ik\eta G \iint_s \vec{s} \times \left[ \vec{s} \times \left( \vec{n} \times \vec{H}_i \right) \right] e^{ik\vec{r}\cdot(\vec{i}-\vec{s})} ds \tag{5}$$

For the mono-static scattering of large targets, $\vec{i} = -\vec{s} \perp \vec{h}_i$, and the following vector relations are obtained [33].

$$\vec{s} \times \left(\vec{n} \times \vec{h}_i\right) = \vec{n} \cdot \left(\vec{s} \cdot \vec{h}_i\right) - \vec{h}_i \cdot (\vec{s} \cdot \vec{n}) = -\vec{h}_i \cdot (\vec{s} \cdot \vec{n}) = \vec{h}_i \cdot \left(\vec{i} \cdot \vec{n}\right) \tag{6}$$

$$\vec{s} \times \vec{h}_i = -\vec{i} \times \vec{h}_i = \vec{e}_i = -\vec{e}_s \tag{7}$$

Therefore, the scattering expression can be written as

$$\vec{E}^s = -2ikH_0\eta G\vec{e}_i \iint\limits_s \left(\vec{i} \cdot \vec{n}\right) e^{2ik\vec{r}\cdot\vec{i}} ds \tag{8}$$

where $\vec{i} \cdot \vec{n} = -\cos(\alpha)$, $\vec{r} \cdot \vec{i} = r$, $r = r_0 + \Delta r$, $\Delta r$ is the changed distance of the scattering point in the integral. In the far field, $r \approx r_0$, and the scattering expression can be expressed as

$$\vec{E}^s = 2ikH_0\eta G\vec{e}_i e^{2ikr_0} \iint\limits_s \cos(\alpha) e^{2ik\Delta r} ds \tag{9}$$

*2.2. Acoustic Scattering of Hard/Soft-Targets*

As shown in Figure 2, the scattering acoustic field $\Phi^s$ can be expressed by the Kirchhoff integral equation [32,34,35]:

$$\Phi^s(r_2) = \frac{1}{4\pi} \iint\limits_s \left[\Phi \frac{\partial}{\partial n}\left(\frac{e^{ikr_2}}{r_2}\right) - \frac{\partial \Phi}{\partial n} \frac{e^{ikr_2}}{r_2}\right] ds \tag{10}$$

where $r_2$ is the distance between the receiver point $M_2$ and the scattering point $d_s$, $n$ is the outward-directed normal, and $\Phi$ is the acoustic potential function on the surface. $M_1$ is the source point, and the radiation field is $\Phi^i = Ae^{ikr_1}/r_1$. Since the Kirchhoff model relies on the approximation that the shadow region does not contribute to the scattering, the common bright region $S_1$ of $M_1$ and $M_2$ is mainly calculated in the scattering calculation [36,37]. For the rigid targets, the acoustic potential meets the following conditions [34,38,39]

$$v_n = \frac{\partial \Phi}{\partial n} = 0 \tag{11}$$

$$\Phi^i + \Phi^s = \Phi \tag{12}$$

where $v_n = v_n^s + v_n^i$, $v_n^i$, and $v_n^s$ are the particle velocity of the incident wave and scattering wave on the boundary in the outward normal direction. On the boundary of the scatterer, the particle velocity generated by the incident acoustic wave is $v_r = -\partial \Phi^i / \partial r$ [3,35,40]. Therefore, $v_n^i$ can be expressed as

$$v_n^i = v_r \cdot (n \cdot r) = -A \frac{ikr_1 - 1}{r_1^2} e^{ikr_1} \cos \alpha_i \tag{13}$$

$$\frac{\partial}{\partial n}\left(\frac{e^{ikr_2}}{r_2}\right) = \frac{ikr_2 - 1}{r_2^2} e^{ikr_2} \frac{\partial r_2}{\partial n} \approx -\frac{ik}{r_2} e^{ikr_2} \cos \alpha_r. \tag{14}$$

Therefore, the acoustic scattering expression $\Phi^s(r_2)$ can be expressed as:

$$\Phi^s(r_2) = -\frac{ikA}{4\pi} \iint\limits_s \frac{e^{ik(r_1+r_2)}}{r_1 r_2} (\cos \alpha_i + \cos \alpha_r) ds. \tag{15}$$

For the mono-static acoustic scattering, $\alpha = \alpha_i = \alpha_r$ and $r_1 = r_2 = r$, such that

$$\Phi^s(r) = -\frac{ikA}{2\pi} \iint\limits_s \frac{e^{2ikr}}{r^2} \cos \alpha ds. \tag{16}$$

In the far field, $r = r_0 + \Delta r$ and $r \approx r_0$, where $\Delta r$ is the change in the position of the integration unit. Therefore, the mono-static velocity potential function of the target is

$$\Phi^s(r) = -\frac{ik\Phi_0^i e^{ikr_0}}{2\pi r_0} \iint_s \cos \alpha e^{2ik \cdot \Delta r} ds \tag{17}$$

where $\Phi_0^i = Ae^{ikr_0}/r_0$ . Similarly, the mono-static velocity potential scattering function for a large soft target can further be expressed as

$$\Phi^s(r) = \frac{ik\,\Phi_0^i e^{ikr_0}}{2\pi r_0} \iint_s \cos \alpha e^{2ik \cdot \Delta r} ds. \tag{18}$$

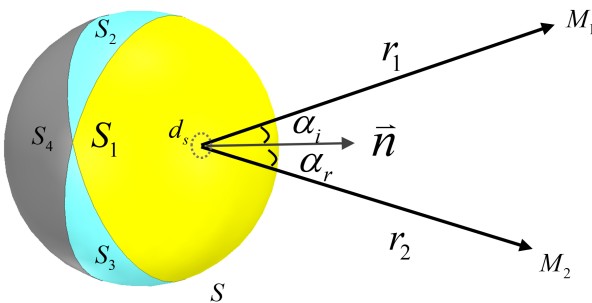

**Figure 2.** Bistatic coordinate system of acoustic scattering.

*2.3. Similarity Relationship and Conditions*

By comparing the EM scattering Equation (9) with the acoustic scattering Equation (17), one can find that the scattering equation is almost identical for both waves. In order to eliminate the influence of the coefficients in the similarity analysis of two kinds of scattering, two scattering expressions were normalized as follows.

$$\text{Norm.}\left(\left|\vec{E}^s(r,\theta)\right|\right) = \text{Norm.}(|\Phi^s(r,\theta)|) = \text{Norm.}\left(\left|\iint_s \cos \alpha e^{2ik \cdot \Delta r} ds\right|_\theta\right) \tag{19}$$

where $\theta$ is the azimuth angle of the incident wave. From the above equation, it can be seen that the normalized EM scattering is described by the same formula as the acoustic scattering equation under the mono-static condition. Under the far-field condition, when the target has the same shape and scale, its normalized scattering characteristics are the same. Therefore, the conditions for the similarity relation between the EM scattering of the conductor and the acoustic scattering of the corresponding rigid conductor are obtained as follows.

$$k_e D_e = k_p D_p \tag{20}$$

where $k_e = 2\pi/\lambda_e$ and $k_p = 2\pi/\lambda_p$ are the wave numbers of the EM wave and the acoustic wave, $D_e$ and $D_p$ are the size of scatterers in the analysis of the EM and acoustic scattering, and $\lambda_e$ and $\lambda_p$ are the wavelength of the EM wave and the acoustic wave. For instance, $D_e = D_p$ means the same size models are used in the analysis of acoustic and EM scattering. Therefore, the constraint can be further expressed as

$$D_e/\lambda_e = D_p/\lambda_p \tag{21}$$

The above equation illustrates that when the ratio of the target size to wavelength is the same, the EM scattering of the 3D conductor target is similar to the acoustic scattering characteristics of the corresponding 3D soft/rigid target.

## 3. Analysis of EM Scattering of 3D Dielectric Targets and Acoustic Scattering of Elastomeric Targets

### 3.1. EM Scattering of 3D Dielectric Targets

According to the physical optics method, as shown in Figure 3, there is a relationship between the incident and the scattered wave for large dielectric targets [33]. Where $\alpha_i$ is the incident angle of the incident wave, and $\hat{e}_\parallel^i$ and $\hat{e}_\parallel^r$ are the vector directions of the incident and reflection electric fields of horizontally polarized waves. For vertically polarized incident waves, $\hat{e}_\perp$ is the vector direction of the incident and reflection electric fields. According to the local coordinate system, as shown in Figure 2, the following relationship can be obtained [41].

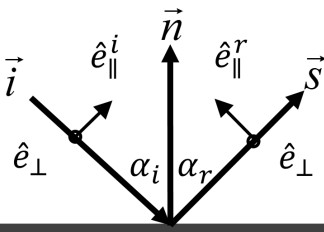

**Figure 3.** The local coordinate system on the boundary of the dielectric target.

$$\hat{e}_\perp = \frac{\vec{i} \times \hat{n}}{\left|\vec{i} \times \hat{n}\right|} \tag{22}$$

$$\hat{e}_\parallel^i = \hat{e}_\perp \times \vec{i} \tag{23}$$

$$\hat{e}_\parallel^r = \hat{e}_\perp \times \vec{s} \tag{24}$$

$\Gamma_\perp$ and $\Gamma_\parallel$ are the reflectivities of the horizontal and vertical polarization wave. Therefore, the incident and scattered electric fields can be written as [33,41,42]

$$\vec{E}^i = \vec{E}_\parallel^i + \vec{E}_\perp^i = E_\parallel^i \hat{e}_\parallel^i + E_\perp^i \hat{e}_\perp \tag{25}$$

$$\vec{E}^s = \vec{E}_\parallel^s + \vec{E}_\perp^s = \Gamma_\parallel E_\parallel^i \hat{e}_\parallel^i + \Gamma_\perp E_\perp^i \hat{e}_\perp \tag{26}$$

where

$$\vec{E}^i\left(r'\right) = \vec{e}_i E_0 e^{j\vec{i}\cdot r'} \tag{27}$$

$$\vec{E}^i\left(r'\right) = \frac{1}{\eta}\vec{i} \times \vec{E}^i\left(r'\right) \tag{28}$$

$$\vec{E}^S\left(r'\right) = \frac{1}{\eta}\vec{s} \times \vec{E}^S\left(r'\right) \tag{29}$$

$$\vec{n} \times \vec{E}^T = (1 + \Gamma_\perp)E_\perp^i(\vec{n} \times \hat{e}_\perp) - \left(1 - \Gamma_\parallel\right)E_\parallel^i \cos\alpha_i \hat{e}_\perp \tag{30}$$

$$\vec{n} \times \vec{E}^T = \frac{1}{\eta}\left[(1 - \Gamma_\perp)E_\perp^i \cos\alpha_i \hat{e}_\perp + \left(1 + \Gamma_\parallel\right)E_\parallel^i(\vec{n} \times \hat{e}_\perp)\right]. \tag{31}$$

Therefore, the scattered electric field can be written as

$$\vec{E}^s = ikG \iint_s \vec{s} \times \left\{ \begin{array}{l} (1 + \Gamma_\perp)E_\perp^i(\vec{n} \times \hat{e}_\perp) - \left(1 - \Gamma_\parallel\right)E_\parallel^i \cos\alpha_i \hat{e}_\perp - \\ \vec{s} \times \left[(1 - \Gamma_\perp)E_\perp^i \cos\alpha_i \hat{e}_\perp + \left(1 + \Gamma_\parallel\right)E_\parallel^i(\vec{n} \times \hat{e}_\perp)\right] \end{array} \right\} e^{ik\vec{r}'\cdot\left(\vec{i}-\vec{s}\right)} ds \tag{32}$$

For the mono-static scattering of large targets, $\vec{i} = -\vec{s}$, $\hat{e}_{\parallel}^i = -\hat{e}_{\parallel}^r$, and $\vec{s} \times \hat{e}_{\perp} = \hat{e}_{\parallel}^i = -\hat{e}_{\parallel}^r$ [42]. The relationships of each direction vector can be written as

$$\vec{s} \times (\vec{n} \times \hat{e}_{\perp}) = (\vec{s} \cdot \hat{e}_{\perp}) \cdot \vec{n} - (\vec{s} \cdot \vec{n}) \cdot \hat{e}_{\perp} = -\cos\alpha_i \cdot \hat{e}_{\perp} \tag{33}$$

$$\vec{s} \times (\vec{s} \times \hat{e}_{\perp}) = (\vec{s} \cdot \hat{e}_{\perp}) \cdot \vec{s} - (\vec{s} \cdot \vec{s}) \cdot \hat{e}_{\perp} = -\hat{e}_{\perp} \tag{34}$$

Therefore, the scattering electric can be further written as

$$\vec{E}^s = 2ikG \iint\limits_{s} \left[ \Gamma_{\parallel} E_{\parallel}^i \hat{e}_{\parallel}^i - \Gamma_{\perp} E_{\perp}^i \hat{e}_{\perp} \right] \cos\alpha_i e^{2ik\vec{r}' \cdot \vec{i}} ds. \tag{35}$$

In the far field, $r = r_0 + \Delta r$ and $r \approx r_0$, where $\Delta r$ is the change in the position of the integration unit. The scattered electric fields can also be reformulated as

$$\vec{E}^s = -2ikGe^{2ikr_0} \iint\limits_{s} \left[ \Gamma_{\perp} E_{\perp}^i \hat{e}_{\perp} - \Gamma_{\parallel} E_{\parallel}^i \hat{e}_{\parallel}^i \right] \cos\alpha_i e^{2ik\Delta r} ds \tag{36}$$

$$= -2ikGe^{2ikr_0} \iint\limits_{s} \left[ \Gamma_{\perp} \left( E_{\perp}^i \hat{e}_{\perp} + E_{\parallel}^i \hat{e}_{\parallel}^i \right) - \left( \Gamma_{\perp} + \Gamma_{\parallel} \right) E_{\parallel}^i \hat{e}_{\parallel}^i \right] \cos\alpha_i e^{2ik\Delta r} ds \tag{37}$$

$$= -2ikGe^{2ikr_0} \iint\limits_{s} \Gamma_{\perp} \left( E_{\parallel}^i \hat{e}_{\parallel}^i + E_{\perp}^i \hat{e}_{\perp} \right) \cos\alpha_i e^{2ik\Delta r} ds + o\left( \vec{E}^s \right) \tag{38}$$

$$\approx -2ikGe^{2ikr_0} \iint\limits_{s} \Gamma_{\perp} \vec{E}^i \cos\alpha_i e^{2ik\Delta r} ds \tag{39}$$

where $o\left( \vec{E}^s \right) = 2ikGe^{2ikr_0} \iint_{s} \left( \Gamma_{\parallel} + \Gamma_{\perp} \right) E_{\parallel}^i \hat{e}_{\parallel}^i \cos\alpha_i e^{2ik\Delta r} ds$. For conductor targets, the reflectivity is $\Gamma_{\perp} = -\Gamma_{\parallel} = -1$. Substituted into the above equation, the same scattering expression as the conductor targets can be obtained, which proves the correctness of the derivation of the above equation.

*3.2. Acoustic Scattering of Elastics*

Since Kirchhoff approximates the common bright regions, the $S_1$ of $M_1$ and $M_2$ is mainly calculated in the scattering calculation. The boundary conditions can be expressed as [40,43,44]

$$\Phi^s = \Gamma_p(\alpha_i)\Phi^i \tag{40}$$

$$\frac{i\omega\rho_1 \left( \Phi^s + \Phi^i \right)}{\partial \left( \Phi^s + \Phi^i \right)/\partial n} = -Z_n \tag{41}$$

where $\Gamma_p(\alpha_i)$ is the acoustic reflectivity [4,26] and $Z_n = \rho_2 c_2 / \cos\alpha_t$ is the acoustic impedance on the surface. The scattered acoustic wave can be written as

$$\Phi^s(r_2) = \frac{1}{4\pi} \iint\limits_{s} \left\{ \Gamma_p(\alpha_i)\Phi^i \frac{\partial}{\partial n} \left( \frac{e^{ikr_2}}{r_2} \right) + \left[ \frac{i\omega\rho_1}{Z_n} \left( 1 + \Gamma_p(\alpha_i) \right)\Phi^i + \frac{\partial\Phi^i}{\partial n} \right] \frac{e^{ikr_2}}{r_2} \right\} ds \tag{42}$$

where [18]

$$\frac{\partial}{\partial n} \left( \frac{e^{ikr_2}}{r_2} \right) = \frac{ikr_2 - 1}{r_2^2} e^{ikr_2} \frac{\partial r_2}{\partial n} \approx -\frac{ik}{r_2} e^{ikr_2} \cos\alpha_r \tag{43}$$

$$\frac{\partial \Phi^i}{\partial n} = \frac{\partial A \frac{e^{ikr_1}}{r_1}}{\partial n} \approx -\frac{ikA}{r_1} e^{ikr_1} \cos \alpha_i. \tag{44}$$

For large targets, the local scattering can be treated as a plane wave so that the reflectivity of the plane wave can be expressed in relation to the surface acoustic impedance as follows [35,44]

$$\frac{\rho_1 c_1 / \cos \alpha_i}{Z_n} = \frac{1 - \Gamma_p(\alpha_i)}{1 + \Gamma_p(\alpha_i)}. \tag{45}$$

Therefore, the scattered acoustic wave can be written as

$$\Phi^s(r_2) = -\frac{ikA}{4\pi} \iint_s \frac{e^{ik(r_1+r_2)}}{r_1 r_2} \Gamma_p(\alpha_i)(\cos \alpha_r + \cos \alpha_i) ds. \tag{46}$$

For the mono-static scattering, $\alpha = \alpha_i = \alpha_r$ and $r_1 = r_2 = r$, and in the far field, $r = r_0 + \Delta r$ and $r \approx r_0$, where $\Delta r$ is the change in position of the integration unit. Therefore, the mono-static velocity potential function of the targets could also be reformulated as

$$\Phi^s(r_2) = -2ik\Phi_0^i G e^{i2kr_0} \iint_s \Gamma_p(\alpha_i) \cos \alpha_i e^{2ik\cdot\Delta r} ds. \tag{47}$$

Comparing the scattering integral equation of a rigid body with that of a general elastomer, it can be seen that there is only one more reflectivity $\Gamma_p(\alpha_i)$ in the integral equation. According to the Huygens principle, the scattered field is generated by the radiation of secondary sources excited by the incident field on the boundary. The potential function of the secondary source is equal to the potential function of the incident wave multiplied by the reflectivity. This means that the secondary potential function is equal to or inverse to the potential function of the incident wave for rigid and soft targets, respectively. Substituting the reflectivity $\Gamma_p(\alpha_i) = 1$ or $\Gamma_p(\alpha_i) = -1$ into expression (47), the same scattering expression can be obtained as (17) and (18), which proves the correctness of the derivation of the above equation.

### 3.3. Similarity Relationship and Conditions

Comparing scattering Equations (39) and (47), one can note that similar expressions exist between the dielectric EM scattering and the corresponding elastic acoustic scattering. When the boundary conditions $|\Gamma_\perp(\alpha_i)| = |\Gamma(\alpha_i)|$ and the scale-frequency conditions $D_e/\lambda_e = D_p/\lambda_p$ are satisfied, the similar relationship between the EM scattering and acoustic scattering can be written as

$$Norm.\left|\vec{E}^s(r,\theta)\right| \approx Norm.|\Phi^s(r,\theta)| = Norm.\left(\left|\iint_s \Gamma_p(\alpha_i) \cos \alpha_1 e^{2ik\cdot\Delta r} ds\right|_\theta\right). \tag{48}$$

where $\theta$ is the incident angle of the EM wave and the acoustic wave relative to the scatterer. The reflectivity of the EM wave $\Gamma_\perp$ and the acoustic wave $\Gamma_p$ can be expressed as [4,45,46]

$$\Gamma_\perp(\alpha_i) = \frac{\cos \alpha_i - \sqrt{(\eta_1/\eta_2)^2 - \sin^2 \alpha_i}}{\cos \alpha_i + \sqrt{(\eta_1/\eta_2)^2 - \sin^2 \alpha_i}} \tag{49}$$

$$\Gamma_p(\alpha_i) = \frac{\cos \alpha_i - \sqrt{(Z_1/Z_2)^2 - \sin^2 \alpha_i}}{\cos \alpha_i + \sqrt{(Z_1/Z_2)^2 - \sin^2 \alpha_i}} \tag{50}$$

where $Z_1$ and $Z_2$ are the acoustic impedance of the media and the scatterer [4,43]. Therefore, the boundary constraint can also be written as $\eta_1/\eta_2 = Z_1/Z_2$, which means that the impedance ratio of the EM wave is equal to that of the acoustic wave. Therefore, the similarity conditions can be proposed as follows.

$$\eta_1/\eta_2 = Z_1/Z_2 \tag{51}$$

$$D_e/\lambda_e = D_p/\lambda_p \tag{52}$$

## 4. Verification and Discussion

Further, to verify the similarity theory of EM scattering and acoustic scattering proposed in this paper, the EM and acoustic scattering of three typical targets are measured, simulated, and analyzed. The EM scattering is measured in the anechoic chamber and simulated by HFSS software. The acoustic scattering is tested in the anechoic tank and simulated by the multi-physics field simulation software COMSOL. The same size and shape are used in measurement and simulation, and $D_e = D_p$, according to Equations (21) and (52), and the wavelengths of the EM wave and underwater acoustic wave are equal. Therefore, when the frequency of acoustic scattering is analyzed at 20 kHz, 4 GHz should be selected for the EM wave to have the same wavelength as the selected acoustic wave, where $\lambda_e = \lambda_p = 75$ mm. In order to quantitatively determine the similarity between EM scattering and acoustic scattering, the correlation coefficient is introduced as follows [47] .

$$r_s = \frac{\sum_{i=1}^{n}(X_i - \bar{X})(Y_i - \bar{Y})}{\sqrt{\sum_{i=1}^{n}(X_i - \bar{X})^2}\sqrt{\sum_{i=1}^{n}(Y_i - \bar{Y})^2}} \tag{53}$$

where $X_i$ and $Y_i$ correspond to the measured amplitude of EM and acoustic scattering at different angles, respectively, and $\bar{X}$ and $\bar{X}$ denote the mean values of EM and acoustic scattering, respectively, in the evaluated angle range. Good correlation and high correlation can be identified when $r_s$ is in the range of $0.7-0.9$ and above 0.9, respectively.

### 4.1. Similarity Verification of EM Scattering by Conductors and Acoustic Scattering by Soft/Hard Targets

Figure 4 shows three typical targets used in the EM and acoustic scattering simulation and measurement. Figure 4a is a metal plate that can be used as a rigid body in the measurement of acoustic scattering. Because the wave impedance of resin material is similar to that of water, as shown in Figure 4b,c, the aircraft and submarine models with a thin resin shell can be selected as soft targets. In the measurement of EM scattering, as shown in Figure 4b,c, the targets are covered with metal, i.e., an electrically conducting material.

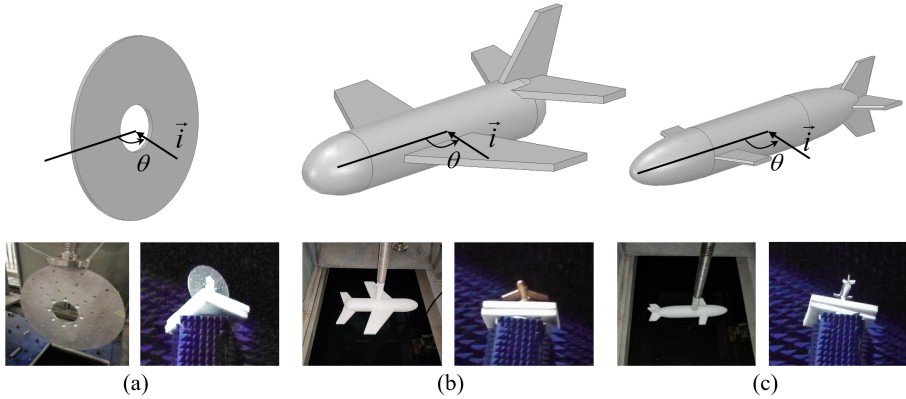

(a)             (b)             (c)

**Figure 4.** Three typical targets selected in the similarity analysis of the EM and acoustic waves. (**a**) Plate. (**b**) Aircraft model. (**c**) Submarine model.

The simulated and measured results of EM and acoustic scattering of a metal plate, an aircraft model, and a submarine model are shown in Figure 5. As shown in Figure 5a,c,e, the simulated EM scatterings of three typical targets agree well with the simulated acoustic scattering. The simulated scattering curves of the main lobe almost overlap, and the trend of change and zero-polarity angles are almost the same. In other words, two simulated scattering curves almost coincide. According to Equation (53), as shown in Table 1, the correlation coefficients of simulated EM and acoustic scattering results are above 0.98, indicating a high similarity.

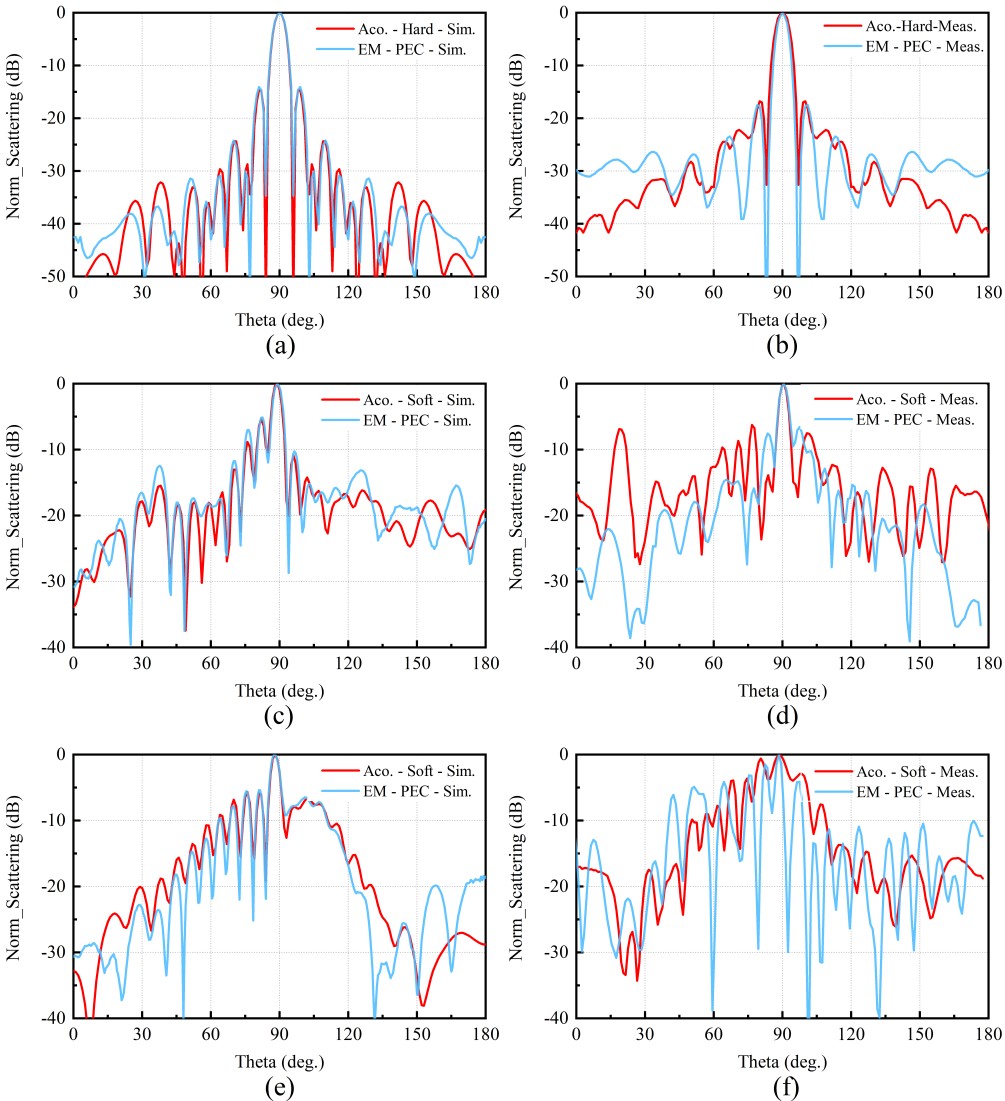

**Figure 5.** Simulated and measured results of EM scattering by conductors and acoustic scattering by hard/soft targets. Scattering simulation results (**a**) and measurement results (**b**) of the plate. Scattering simulation results (**c**) and measurement results (**d**) of the aircraft model. Scattering simulation results (**e**) and measurement results (**f**) of the submarine model.

As shown in Figure 5b,d, the measured EM scattering curves of the plate and aircraft agree well with the measured acoustic scattering. Additionally, the measured and simulated results are in good agreement. Therefore, the scattering curves of the main lobe almost overlap, and the trend of change and zero-polarity angles also have good agreement. The correlation coefficients of measured EM and acoustic scattering results are 0.97 and 0.91 are highly similar.

Compared with the simulation results, there are many differences in the measured EM and acoustic scattering results of the submarine. Even though the details of the scattering curves are not good enough, the trends of change in the two measured scattering curves are similar, with a correlation coefficient of 0.65. This is related to the smaller size of the submarine model, and the weaker scattered signal is easily affected in the EM and acoustic scattering measurement.

In general, based on the above similarity analysis and a comparison with the target with a complicated structure and small size, a better similarity characteristic can be obtained with a relatively simple structure and larger size target, and the simulated scattering curves are better than the measurement results.

**Table 1.** Correlation coefficients of simulated and measured EM and acoustic scattering results with different boundary conditions.

| Targets | $\Gamma = 0.1$ | $\Gamma = 0.5$ | $\Gamma = 0.8$ | $\|\Gamma\| = 1$ [1] | $\|\Gamma\| = 1$ (meas.) |
|---|---|---|---|---|---|
| Plate | 0.99 | 0.99 | 0.99 | 0.99 | 0.98 |
| Aircraft | 0.93 | 0.98 | 0.99 | 0.99 | 0.91 |
| Submarine | 0.97 | 0.97 | 0.96 | 0.97 | 0.65 |

[1] $\|\Gamma\| = 1$ means the reflectivity of the EM wave of the conductor and acoustic wave of the hard/soft target.

### 4.2. Similarity Verification of the EM Scattering by Dielectrics and Acoustic Scattering by Elastics

Due to the major difficulty in the fabrication of impedance materials, only the simulation result is used in the similarity verification of EM scattering by dielectrics and acoustic scattering by elastomers. As shown in Figure 4 and Table 1, the reflectivity is set to 0.1, 0.5, and 0.8, which corresponds to impedance ratios $\eta_1 / \eta_2 = Z_1 / Z_2 = 9/11$, $1/3$, and $1/9$. The simulation results of EM scattering and acoustic scattering with different boundary conditions are shown in Figure 6. It can be found that all of the EM scattering and acoustic scattering results have a very good agreement with different boundary impedances. The maximum scattering direction, the width of the main lobe, and the two scattering curves have a perfect agreement. The correlation coefficients of the two scattering curves are greater than 0.96 and are highly correlated. Furthermore, two kinds of scattering curves are more consistent with a more simple structure, larger aperture, and higher reflectivity. Note that different software is used in the simulation of EM scattering and acoustic scattering. Above all, the results in Figure 6 illustrate similar characteristics of EM and acoustic scattering with different boundary conditions.

The above simulated and measured results verify the proposed similarity theory of EM and acoustic scattering. This illustrates that the acoustic scattering characteristic of the target can be simulated by the EM scattering under certain boundary conditions and vice versa.

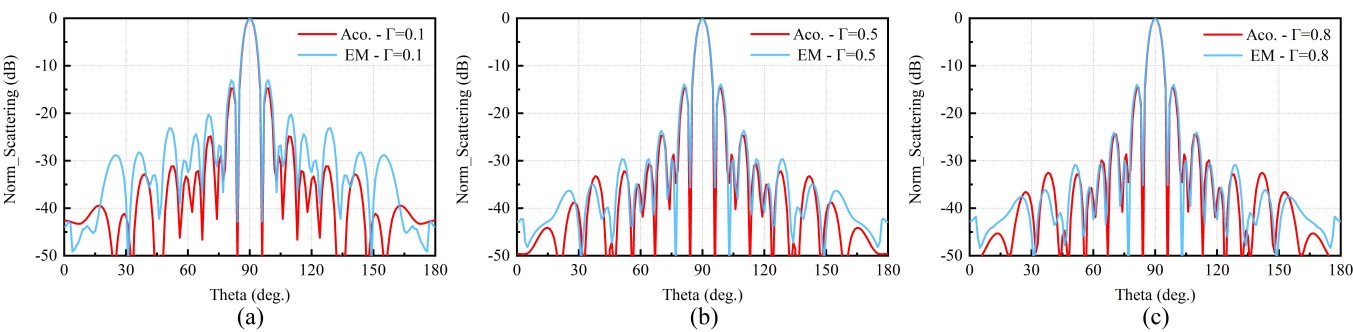

**Figure 6.** *Cont.*

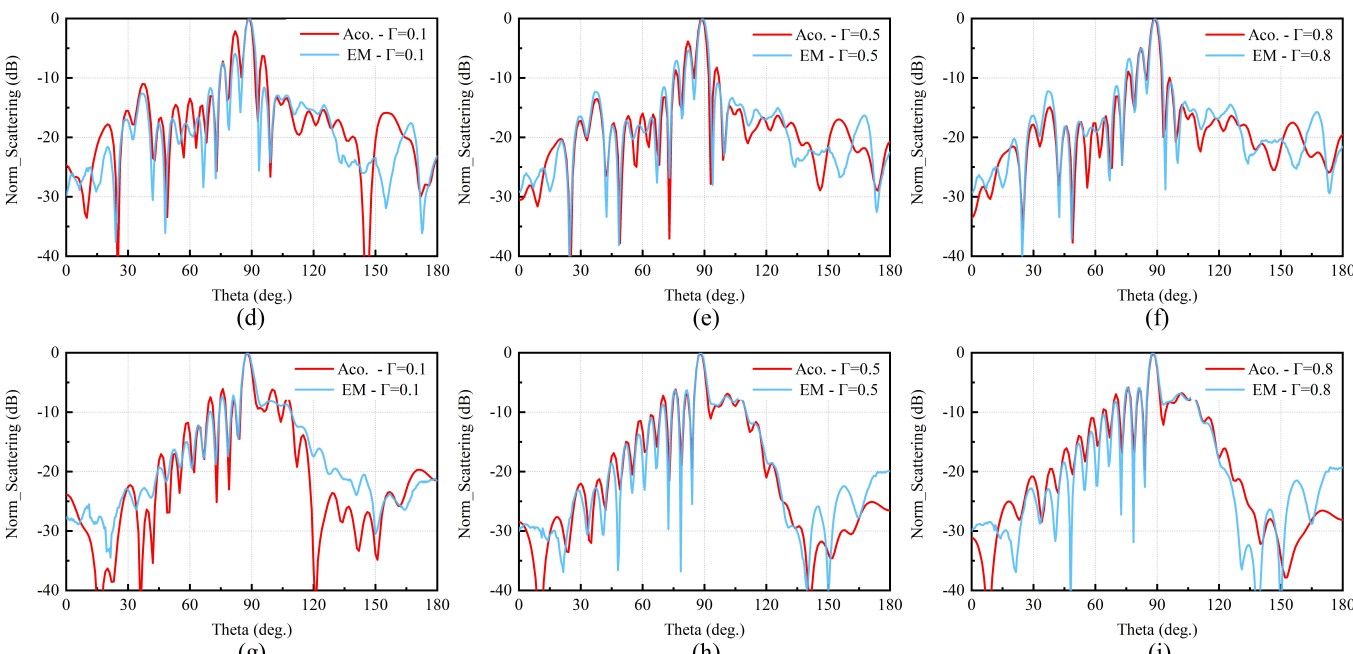

**Figure 6.** Simulated results of EM and acoustic scattering with different boundary conditions. (**a**–**c**) Scattering simulation results of the plate. (**d**–**f**) Scattering simulation results of the aircraft model. (**g**–**i**) Scattering simulation results of the submarine model.

## 5. Conclusions

In this work, based on the method of physical optics, the similarity between the EM scattering by a 3D conductor and the acoustic scattering by a corresponding 3D hard/soft target is studied. Further, the similarity between the EM scattering by a dielectric material and the acoustic scattering by the corresponding elastic material is also developed. Additionally, the similarity conditions of EM and acoustic scattering are derived. According to the proposed conditions, based on different simulation software, the EM and acoustic scattering of three typical targets with different reflectivities are simulated and analyzed. The simulated results indicate a good agreement between EM scattering and acoustic scattering within the range of verified angles, and the correlation coefficients are above 0.93, indicating a high correlation. Moreover, the EM scattering and acoustic scattering are tested in an anechoic chamber and an anechoic tank, respectively. Although idealized boundary conditions are considered here, the measured EM scattering curves of the plate and the aircraft agree well with the measured acoustic scattering results. At the same time, the measured and simulated results are in good agreement. For the smaller scatterer employed (the submarine model), the distribution and magnitude of poles and zeros do not match sufficiently, but the overall trends of the measured EM and acoustic scattering patterns are similar, with a correlation coefficient of 0.65. Note that different software is used in the simulation of EM scattering and acoustic scattering. The good scattering similarity, both in simulation and measurement, illustrates that the target acoustic scattering can be used to replace the EM scattering characteristics at a sufficient accuracy based on the derived similarity conditions and vice versa. The research results provide a new method and idea for the simulation and measurement of target EM/acoustic scattering and have great significance for the acquisition of scattering characteristics and the structural design of targets. Future work will be directed towards the inclusion of the effects of various other important parameters, such as the relationship between EM and acoustic scattering intensity, 1D and 2D range profiles, and scattering characteristics of targets with some stealth ability and composite material. In addition, because of the boundaries of oceanic waveguides [9], the problem of underwater acoustics scattering by AUVs, submarines, and ships is more

complex, and the similarity between the EM scattering and the acoustic scattering by targets in the boundaries space is an important research topic for future studies.

**Author Contributions:** Conceptualization, J.W. and J.G.; methodology, J.W.; software, H.L.; validation, J.W., H.L. H.G and Q.Z.; formal analysis, J.W.; investigation, J.G.; resources, Q.Z.; data curation, J.W.; H.G. and Q.Z.; writing—original draft preparation, J.W.; writing—review and editing, J.W.; visualization, J.W. and H.L.; supervision, J.G.; project administration, J.G.; funding acquisition, J.G. All authors have read and agreed to the published version of the manuscript.

**Funding:** This research was funded in part by the National Natural Science Foundation of China under grant 42027805 and the Jiangsu Innovation and Entrepreneurship Group Talents Plan.

**Institutional Review Board Statement:** Not applicable.

**Informed Consent Statement:** Not applicable.

**Data Availability Statement:** Not applicable.

**Acknowledgments:** The authors greatly appreciate the support provided by the Microwave Lab of NUIST, Nanjing, China.

**Conflicts of Interest:** The authors declare no conflict of interest.

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
