# Peer review of "Similarity Study of Electromagnetic and Underwater Acoustic Scattering by Three-Dimensional Targets in Unbounded Space"

_jmse, doi:10.3390/jmse11020440_

Round 1

Reviewer 1 Report

The manuscript presents the results of the application of the physical optics method for analysis of the similarity between the EM scattering by conductor and the acoustic scattering by target. It is shown that the target acoustic scattering can be used to simulate the EM scattering characteristics under certain boundary conditions and vice versa. The presented results could be of interest for JMSE readers. In general, the manuscript is well-written, and I would support its publication after some improvements. 

Here are few my remarks.

I) Major remark.

I suppose that authors should emphasize one important point. They research is relating to acoustic and EM scattering in unboundaried space. So, I offer to add the phrase “in unboundaried space” in title of manuscript.

Similarity Study of Electromagnetic and Underwater Acoustic Scattering by Three-Dimensional Targets in Unboundaried Space. 

The reason of my offering is the following. In the underwater acoustics the problem of scattering by AUVs, submarines, ships is more complex due to boundaries of oceanic waveguide (water-air and water-bottom boundaries). Authors do not take into account boundaries of space.

I offer add to paper introduction the paragraph about complexity of the problem of acoustic scattering in underwater waveguide and add some references relating to this problem.

1) Sarkissan A. Method of superposition applied to scattering from a target in shallow water //J. Acoust. Soc. Amer. 1994. V. 95. â„– 5. P. 2340-2345. (doi: 10.1121/1.409870)

2) Kuz'kin V.M., Grigor'ev V.A. Field Focusing Control in Multimode Plane-Layered Waveguides // Acoustical Physics. 2005. 51(3). p. 292-299 (doi: 10.1134/1.1922542)

3) Kuz'kin V.M. Sound Diffraction by an Inhomogeneity in an Oceanic Waveguide //  Acoustical Physics. 2002. 48(1). p. 69-75 (doi:10.1134/1.1435392)

The similarity between the EM scattering by conductor and the acoustic scattering by target in boundaried space is very important question. The method offered in paper allows to take into account space boundaries or not? It may be aim of the following research of the authors.  

II) Minor remark.

The references should be numbered in the order of their mentioned in the paper. But firstly authors put references [1,2]. Then they put references [36,39], after  put references [5,6].

Reviewer 2 Report

Physical optics (PO) is a classical approach to determine electromagnetic and acoustic far field scattering patterns. The formal similarity arising in applications of PO in these two physical domains has been considered in several scientific publications. In the present paper, the authors present a very practical and usable view on this similarity and formally derive conditions, under which electromagnetic and acoustic scattering data can be transferred into the other field. They consider the differences for electromagnetic scattering at good conductors (related to hard acoustic scatterers) and at dielectrics (related to elastic acoustic scatterers). A particular merit of this paper are the nice experimental results that both support the theoretical results about the similarity of measurements in both physical domains and show the accuracy of the computation of the scattering results by PO. Hence, I consider this work as a substantial and new scientific result that deserves publication.

However, although the basic ideas and results of the paper are clearly presented, the article should carefully be reviewed before it is considered for publication, because there is still potential to enhance its readability and comprehensibility. In addition, some more information should be given to the reader. In reviewing the text, the authors should address the following points:

1.)   The text needs a careful review. Sometimes, expressions are used, that are misleading or not very common in the relevant context. In other cases, typos aggravate understanding the text. Examples are (here, I only can present some examples – the authors are kindly asked to proactively search for points to be enhanced)

Meanwhile, constraints for scattering similarity relationships are proposed ->  Particularly, conditions under which similar scattering patterns arise are derived (p 1, l 7)

various boundaries -> different type of boundaries (p 1, l 9)

The measured results between the two kinds of scattering -> The correlation coefficients between the electromagnetic and acoustic scattering patterns (p 1 , l 11)

abundant laboratory -> abundant laboratory capacities (p 1, l 19)

the generating -> the generation (p 1, l 20)

in the outfield (?), (p 1, l 28)

is a mechanically kinds -> is a mechanical kind (p 1, l 30)   

circular desk -> circular disk (p 2, l 40)

simulation measurement (?), (p 2, l 40)

victor -> vector (p 3, l 96)

Biatatic -> Bistatic (p4, before l 107)

Since Kirchhoff approximates that -> Since the Kirchhoff model relies on the  approximation that (p 4, l 112) 

scatter -> scatterer (p 4, l 117, and further incidence)

single station -> monostatic (?), (p 5, l 121, and further incidence)

EM scattering has the same -> EM scattering is described by the same (p 5, l 133)

existed -> excited (?), (p 6, l 147)

horizontal polarization wave -> horizontally polarized wave (p 6, l 140)

vertical polarization incident wave -> vertically polarized incident wave (p 6, l 140)

monostation velocity potential function -> monostatic velocity potential function (p 8, l 178)

It is means -> This means (p 8, l 185)

And Substituted -> Substituting (p 8, l 187)

similarity constraints -> similarity conditions (p 8, l 199)

A thickness resin material shells -> a thick resin shell (?), (p 9, l 222)

metal which can be used as conductor -> metal, i.e., an electrically conducting material (p 9, l 224)

good agree -> agree well (p 12, l 282) 

For the smaller size … applied, some details … is not good enough -> For the smaller sized … employed, some details … do not match in a sufficient way (p 12, l 284) [remark: these details should be specified]

But the change trend of two measurement scattering curves are similar (?), (p 12, l 285)

Check also for articles, typos, and missing blanks …

2.)   The formulae in the theoretical part must carefully be reviewed. Among others, this comprises the following points:

Unique variables have to be used throughout the text and all variables have to be properly explained in the text e.g., is D or kD the size of the scattering object? (p 2, l 76), J_s is the current density, not “surface currents” (p 3 , l 100), what are E and H on the right hand side of formulae (1) and (2)?, do not switch from i to j in (14), why is the speed of sound denoted v_n (p 4, l 116) and c_1 in (45) and c_2 in (p 7, l 170)?, etc.

Please use upright fonts for mathematical functions (cos, sin) with a small space before the argument. Please do not use brackets that are not required

It is recommended to separate between indices that are names and that are variables. The first ones should be printed with upright fonts, the latter ones in italics. 

Please give more information about the area over which is integrated. In PO usually the shadowed regions are left out. This is not done in this paper. Are the integrals still in all cases formally correct (particularly, when the incident angle is integrated over)?

Please give more information over which variable the integrations are carried out and which are parameters (distinguish between test point, source point and the difference between these two – they all occur as arguments in different terms in the integrands). Does, e.g., the kernel function (Green’s function) in (10) depend on the test point only (it is constant in this case)? With respect to which variable are the normal derivatives computed? 

Chance O by another abbreviation in (38)

What are r and theta in (48)

How is alpha (p 4, l 104) related to the notions in Fig. 2?

Are s_e and s_p the “scales of the EM wave and acoustic wave” or just the dimension of the objects under test? 

… etc. …

3.)   More information on the experimental part are required, including:

What are the dimensions of the scattering models and the test environment?

Please give further information on the measurement sites.

What is the distance between the models and the source/measurement device?

What is taken as source/measurement device?

Are far field conditions guaranteed? 

Is the 10-wavelength-condition for the size of the objects fulfilled, which was mentioned in the text?

How have the acoustic material parameters for the elastic bodies been determined? What are their values? How has the hardness of the hard body been guaranteed?

Please give all relevant data permit the reader to check the similarity conditions (electromagnetic and acoustic impedances, mass densities, speed of sound etc.)

Please use exact quantitative descriptions on the measurements also in the interpretation of the results

Please give relevant information in the conclusions instead of vague hints, as, e.g., “constraints on the validity” -> “we showed that acoustic measurements can replace EM measurements and vice versa at sufficient accuracy in the case that …”, “some details” (what details?), “certain boundary conditions” (what boundary conditions?) 

Round 2

Reviewer 2 Report

First of all, I wish to thank the authors for the careful review of their paper and the detailed answers to the reviewers’ comments. The authors enhanced the manuscript significantly.

I suggest that the authors revise the paper ones more before it will be published. The authors may find information in the list of items below, which I noticed when reading the text (please note that the list contains only some examples, other points that can be improved could not be listed). 

Titel:

in Unboundaried Space -> in Unbounded Space

Abstract:

The electromagnetic (EM) scattering and underwater acoustic scattering characteristics of 1 targets are important research contents in related fields. However, there are some difficulties in the 2 simulation and measurement scattering by targets with large sizes. So, the similarity study between 3 acoustic and EM scattering provides a reference method for the simulation of scattering characteristics. -> The characterization of targets by electromagnetic (EM) scattering and underwater acoustic scattering is an important object of research contents in these two related fields. However, there are some difficulties in the  simulation and measurement of the scattering by targets with large size. Consequently, a similarity study between acoustic and EM scattering may help to exchange results from one domain into the other and even provide a general reference method for the simulation of scattering characteristics in both fields.

Particularly, conditions under which similar scattering patterns arise are derived -> It is particularly derived, how quantities of one domain have to be transferred into the other so that similar scattering patterns arise

Then, according to the conditions -> Then, according to these transfer rules

verify the proposed similarity theory and conditions of EM and acoustic scattering and -> verify the proposed similarity theory of EM and acoustic scattering including the transfer from one domain into the other and

Introduction:

still remain. For example, -> still remain, for example, 

the speed of EM waves is so fast -> the speed of EM waves is so fast

and unfriendly test environments -> and inconvenient test environments

and EM scattering of circular disk -> and EM scattering of circular disks

In [28], acoustic simulated measurement of radar high-resolution range 57 profile is studied and many merits of underwater acoustic simulation measurement are -> In [28], simulated acoustic data  of high-resolution radar  range profiles are studied and many merits of simulated underwater acoustic data are 

targets with a size D (or k · D ) above 10 times -> targets with a size D with D>20 pi  above 10 times                 (isn’t D/lambda>10 easier for the reader?)

Similarity of EM Scattering by Conductors and Acoustic Scattering by Hard-Targets

Similarly, the mono-station velocity potential -> Similarly, the mono-static velocity potential

Use i instead of j for the imaginary unit in (44)

For the mono-station station scattering -> For the mono-static scattering

4. Verification and Discussion:

wave should be selected at 4GHz for the same wavelength wave the selected acoustic wave -> wave should be selected at 4GHz to have the same wavelength as the selected acoustic wave

The authors sent all data of the measurement that I asked for to me. I would suggest to integrate these data into the paper (maybe in form of a table).

5. Conclusions

between the EM scattering by 3D conductor and the acoustic scattering by corresponding 3D hard/soft target -> between the EM scattering by a 3D conductor and the acoustic scattering by a corresponding 3D hard/soft target

And the similarity of the EM scattering by dielectric and the acoustic scattering of the corresponding elastic -> Further the similarity between the EM scattering by a dielectric material and the acoustic scattering by the corresponding elastic material 

Although in the idealized boundary considered here, the measured EM scattering curves of the plate and aircraft agree well with the measured acoustic scattering results -> Although idealized boundary conditions are considered here, the measured EM scattering curves of the plate and the aircraft agree well with the measured acoustic scattering results
